Summer diatom blooms in the eastern North Pacific gyre investigated with a long-endurance autonomous surface vehicle

Anderson Emily E. 1
Wilson Cara 2
Knap Anthony H. 3
http://orcid.org/0000-0002-5717-8163 Villareal Tracy A. 1 tracyv@austin.utexas.edu
1 Department of Marine Science and Marine Science Institute, The University of Texas at Austin , Port Aransas, TX , USA
2 National Marine Fisheries, National Oceanic and Atmospheric Administration , Monterey, CA , USA
3 Geochemical and Environmental Research Group, Texas A&M University , College Station, TX , USA
Berges John
Electronic publication date: 2018 Aug 15
Publication date: 2018
Volume: 6
Electronic Location ID: e5387
Received 2018 May 24; Accepted 2018 Jul 17
Copyright: © 2018 Anderson et al.
Copyright year: 2018
Copyright holder: Anderson et al.
License: This is an open access article distributed under the terms of the Creative Commons Attribution License, which permits unrestricted use, distribution, reproduction and adaptation in any medium and for any purpose provided that it is properly attributed. For attribution, the original author(s), title, publication source (PeerJ) and either DOI or URL of the article must be cited.
License URL: https://creativecommons.org/licenses/by/4.0/

Keywords: Remote sensing, Diatom bloom, Oligotrophic ocean, Marine, Symbiosis, Phytoplankton, Autonomous vehicle, Diatoms

Funding: Liquid Robotics National Science Foundation award OCE 1430048 and 1537546 This work was funded by Liquid Robotics, a Boeing company, through a cash grant, glider time, and technical assistance. In addition, this work was supported by the National Science Foundation award OCE 1430048 and 1537546. The funders had no role in study design, data collection and analysis, decision to publish, or preparation of the manuscript.

==============================
Satellite chlorophyll a (chl a) observations have repeatedly noted summertime phytoplankton blooms in the North Pacific subtropical gyre (NPSG), a region of open ocean that is far removed from any land-derived or Ekman upwelling nutrient sources. These blooms are dominated by N2-fixing diatom-cyanobacteria associations of the diatom genera Rhizosolenia Brightwell and Hemiaulus Ehrenberg. Their nitrogen fixing endosymbiont, Richelia intracellularis J.A. Schmidt, is hypothesized to be critical to the development of blooms in this nitrogen limited region. However, due to the remote location and unpredictable duration of the summer blooms, prolonged in situ observations are rare outside of the Station ALOHA time-series off of Hawai’i. In summer, 2015, a proof-of-concept mission using the autonomous vehicle, Honey Badger (Wave Glider SV2; Liquid Robotics, a Boeing company, Sunnyvale, CA, USA), collected near-surface (<20 m) observations in the NPSG using hydrographic, meteorological, optical, and imaging sensors designed to focus on phytoplankton abundance, distribution, and physiology of this bloom-forming region. Hemiaulus and Rhizosolenia cell abundance was determined using digital holography for the entire June–November mission. Honey Badger was not able to reach the 30°N subtropical front region where most of the satellite chl a blooms have been observed, but near-real time navigational control allowed it to transect two blooms near 25°N. The two taxa did not co-occur in large numbers, rather the blooms were dominated by either Hemiaulus or Rhizosolenia. The August 2–4, 2015 bloom was comprised of 96% Hemiaulus and the second bloom, August 15–17, 2015, was dominated by Rhizosolenia (75%). The holograms also imaged undisturbed, fragile Hemiaulus aggregates throughout the sampled area at ∼10 L−1. Aggregated Hemiaulus represented the entire observed population at times and had a widespread distribution independent of the summer export pulse, a dominant annual event suggested to be mediated by aggregate fluxes. Aggregate occurrence was not consistent with a density dependent formation mechanism and may represent a natural growth form in undisturbed conditions. The photosynthetic potential index (Fv:Fm) increased from ∼0.4 to ∼0.6 during both blooms indicating a robust, active phytoplankton community in the blooms. The diel pattern of Fv:Fm (nocturnal maximum; diurnal minimum) was consistent with macronutrient limitation throughout the mission with no evidence of Fe-limitation despite the presence of nitrogen fixing diatom-diazotroph assemblages. During the 5-month mission, Honey Badger covered ∼5,690 km (3,070 nautical miles), acquired 9,336 holograms, and reliably transmitted data onshore in near real-time. Software issues developed with the active fluorescence sensor that terminated measurements in early September. Although images were still useful at the end of the mission, fouling of the LISST-Holo optics was considerable, and appeared to be the most significant issue facing deployments of this duration.

Introduction

Low-nutrient, low chlorophyll oceanic regimes with chlorophyll a (chl a) concentrations <0.07 mg m−3 constitute approximately 60% of the world ocean (Guieu et al., 2014) and are home to a phytoplankton community highly adapted for survival at the ambient nanomolar concentrations of inorganic NO3− and PO4−3. One of the important adaptations is nitrogen-fixation (diazotrophy), a process by which dissolved N2 is converted into ammonium for incorporation into amino acids and proteins (Carpenter & Capone, 2008). Dizaotrophy requires abundant iron resources (Mills et al., 2004; Ratten et al., 2015) and is reduced in iron-limited regions. N2-fixation may also be limited by other nutrients (Kustka, Carpenter & Sanudo-Wilhelmy, 2002; Mills et al., 2004; Ratten et al., 2015) or competition by non-diazotrophic phytoplankton (Weber & Deutsch, 2014). Multiple prokaryote taxa are capable of diazotrophy (Zehr & Kudela, 2011); photosynthetic taxa include colonial cyanobacteria such as Trichodesmium spp. (Capone et al., 1997; Goering, Dugdale & Menzel, 1966), free-living coccoid forms including Crocosphaera watsonii (Goebel et al., 2008; Zehr et al., 2001), and coccoid or filamentous forms symbiotic with eukaryotes. Of these latter symbioses, there are coccoid forms symbiotic with the prymnesiophyte Braarudosphaera bigelowii (Gran and Braarud) Deflandre (Thompson et al., 2012, 2014), dinoflagellates (Farnelid et al., 2010; Foster, Carpenter & Bergman, 2006) and filamentous or coccoid cyanobacteria occurring as exo- or endosymbionts of diatoms (Foster & O’Mullan, 2008; Villareal, 1992). This latter group, diatom-diazotroph associations (DDAs), are dominated by an endosymbiosis between the filamentous cyanobacteria, Richelia intracellularis, and members of the diatom genera Rhizosolenia and Hemiaulus. These symbioses have complex interactions with their hosts (Foster & Zehr, 2006; Hilton et al., 2013) and the taxonomic distinctness of the symbionts even within a single host genus remains unclear. DDAs play important roles in biogeochemical cycling off the Amazon (Carpenter et al., 1999; Subramaniam et al., 2008) and Mekong Rivers (Bombar et al., 2011) as well as in the central North Pacific gyre (Church et al., 2008).

At the Hawai’i Ocean Time-series (HOT), episodic pulses of DDAs dominated by Hemiaulus spp. rapidly sink to depth (Scharek et al., 1999; Scharek, Tupas & Karl, 1999) and transport ∼20% of the annual benthic carbon flux in a limited window (July 15–August 15) termed the summer export pulse (Karl et al., 2012). Isotopic signatures of N2 fixation suggest that their diazotrophic symbiont is present and fueling the biomass flux; the rapid sinking rate indicates aggregation plays a key role in the accelerated transport to depth (Scharek, Tupas & Karl, 1999). The summer export pulse is possibly linked to episodic surface blooms of DDAs advecting through the region in the prevailing flow (Dore et al., 2008; Fong et al., 2008; White, Spitz & Letelier, 2007). Auxospore formation has also been offered as an explanation (Karl et al., 2012) although direct examination of trap material (Scharek et al., 1999; Scharek, Tupas & Karl, 1999) reported no evidence of auxosporulation. Follett et al. (2018) modeled generalized diatom-diazotroph association dynamics, noting that the population peaked in the early summer and rapidly declined during the summer export pulse window after a transition from modelled Fe to P limitation favored competitive exclusion by other taxa. The model necessarily addressed generalized conditions and did not address the localized blooms noted by satellites. These blooms dominate in the summer (Wilson, 2003) and are often associated with the unique properties of mesoscale eddy flow-fields (Calil et al., 2011; Calil & Richards, 2010; Guidi et al., 2012). There are few long-term, high frequency direct observations on diatom-diazotroph association abundance to evaluate these hypotheses.

In the North Pacific, the diatom-diazotroph association host genus Hemiaulus is a characteristic upper euphotic zone species typically found across the central North Pacific gyre at concentrations of ∼102 cells L−1 (Venrick, 1988, 1999). Near-surface blooms of both Rhizosolenia and Hemiaulus DDAs at 104 cells L−1 (Venrick, 1974) extend well north of Hawai’i at abundance up to 104 L−1 (Brzezinski, Villareal & Lipschultz, 1998; Krause et al., 2012; Villareal et al., 2011) and are frequently associated with summer chl a blooms observed in satellite ocean color sensors (Villareal et al., 2011). These chl a blooms (operationally defined as > 0.15 mg chl a m−3) north of 25.5°N cover a much greater range of temperatures and surface area than the blooms at HOT (∼22.5°N) and extend at least as far north as 35.5°N (Villareal et al., 2012). While the data suggest that these satellite-observed blooms are probably associated with diatom-diazotroph association events, it has remained difficult to sample these more northerly blooms due to the remote location, episodic timing and extensive geographic range. The applicability of the summer export pulse to these areas is unclear, as is the general role of aggregation in Hemiaulus spp. biology. In situ diver observations suggest aggregation commonly occurs in Hemiaulus (Villareal et al., 2011), providing a means for rapid sinking as the bloom senesces. It is unclear whether Hemiaulus aggregation occurs as a density dependent process as noted in coastal diatom blooms (Burd & Jackson, 2009; Jackson, 2005), is a natural growth form of the genus similar to Rhizosolenia mats, is uniquely localized to the summer export window, or is a more generalized feature throughout the year. With recent observations of the ubiquitous presence of living diatom cells in the 2,000–4,000 m depth strata, the role of aggregation in oceanic diatom biology has assumed new importance (Agusti et al., 2015).

Sampling these blooms outside of HOT is a challenge due to both the distance to blooms, unpredictable occurrence, long planning lead time, and cost involved in multiple week research cruises. Even at HOT, shipboard sampling is at approximately monthly intervals and insufficient to resolve episodic events in annual cycles. To address this, we used an SV2 Wave Glider (Honey Badger), a long-range autonomous vehicle utilizing wave power for propulsion and solar panel arrays on a surface float to provide power for a variety of sampling instruments (Daniel, Manley & Trenaman, 2011). While many types of autonomous vehicles are used in the marine environment (Dickey et al., 2008; Lee et al., 2017), the Wave Glider is particularly capable of multiple-month missions carrying extensive payloads, is under near-real time control, and has successfully transited from Hawai’i to Australia while returning oceanographic data (Villareal & Wilson, 2014). They have been successfully deployed for sediment transport studies (Van Lancker & Baeye, 2015), wind/current assessments of typhoons (Van Lancker & Baeye, 2015), buoy validation exercises (Fitzpatrick et al., 2015), examination of air–sea coupling in the Southern Ocean (Thomson & Girton, 2017), and processes controlling North Atlantic and Eastern Pacific Ocean salinity variability (Lindstrom et al., 2017).

In our study, we equipped the Wave Glider Honey Badger with a novel array of imaging and photophysiology sensors specifically targeting phytoplankton dynamics. We present data gathered during a 5-month mission in 2015 which sampled two diatom blooms. The mission objectives were to return the glider after 5 months with all sensors collecting useful data, determine if a holographic imaging system could quantify diatom events, relate the abundance to satellite observed chl a blooms, examine the data for Hemiaulus aggregations, and acquire photosynthetic efficiency data using active fluorescence.

Materials and Methods

The mission area for the Honey Badger was the eastern North Pacific subtropical gyre (NPSG) spanning 19–30°N and 144–157°W in the open waters northeast of the Hawaiian Islands (Fig. 1) where chl a blooms regularly occur between July and October (Wilson, 2003). Waypoints were chosen based on Aqua-MODIS 8-day composite chl a concentration satellite images from the Environmental Research Division’s ERDDAP (https://coastwatch.pfeg.noaa.gov/erddap/griddap/erdMBchla8day.html). After a preliminary deployment in the test area off Kawaihae, Hawai’i, the Honey Badger headed north on June 1, 2015. It was recovered on November 3, 2015 and returned to the test facility for evaluation and data download.

Figure 1 Mission track of the SV2 Wave Glider Honey Badger.

Mid-day positions points are color-coded by month. The asterisk north of Oahu is Station ALOHA of the Hawai’i Ocean Time-Series (HOT).

The Wave Glider® SV2 (Liquid Robotics, a Boeing company, Sunnyvale, CA, USA) is an autonomous surface vehicle capable of extended operations offshore. It has a surface float (2.1 × 0.6 m) connected by an umbilical (seven m in this application) to a subsurface glider (0.4 × 1.9 m) with articulating wings (1.1 m wide) that uses vertical motion from waves to provide forward movement. Within the surface float, equipment bays provide space for computers, communications equipment and battery arrays powered by solar panels. Iridium satellite communication with the Wave Glider Honey Badger used in this mission was in near-real time and provided a near immediate ability to course correct and respond to environmental conditions.

The Honey Badger was equipped with sensors on the float, the sub-body, and on a towed body (Fig. 2; Table 1). The float contained two Turner Designs C3 fluorometers (Turner Designs, Sunnyvale, CA, USA) rimmed with anti-fouling copper, a Seabird Electronics gpCTD (Seabird Electronics, Inc., Bellevue, WA, USA) for water temperature and salinity with an inline antifouling tablet, a Canon G10 camera (Canon, USA Inc., Melville, NY, USA) looking down through the float, a Datawell MOSE weather sensor (Datawell BV, Haarlem, The Netherlands), Airmar WX and WS weather sensors+light bar (Airmar Technology Corporation, Milford, NH, USA), an automatic identification system (AIS) transponder, and a radar reflector. The sub-body located seven m below the float had an externally mounted Turner Designs PhytoFlash (Turner Designs, Sunnyvale, CA, USA) utilizing the data and power connections through the umbilical. The PhytoFlash sensor was shielded by a dark cap painted inside and out with anti-fouling paint. A Sequoia Scientific, Inc. (Bellevue, WA, USA) Laser in situ Scattering and Transmissometry Holographic System (LISST-Holo, termed Holo) was deployed in a neutrally buoyant towed body behind the Honey Badger on a 10 m tether equipped with scoops to passively direct water into the sample field. The tow fish varied from 6.3 to 15.5 m deep based on the Holo’s internal depth sensor. The Holo drew power from the umbilical with the data stored in the Holo’s onboard internal memory module. Bandwidth limitations did not permit transmission to shore via Iridium satellite. The Holo sample chamber was painted with antifouling paint and lined with copper tape on other surfaces to minimize fouling. Power consumption and available solar charging dictated sampling frequency and varied with the sensors (Table 1). The vehicle reported location and condition telemetry every 30 s. Sensors were integrated into the onboard processing and communications equipment by Liquid Robotics with the exception of the PhytoFlash. Software integration for the PhytoFlash and tow body construction was provided by the Geophysical Engineering Research Group (GERG) at Texas A&M University.

Figure 2 Honey Badger diagram and sensor locations.

Schematic provided courtesy of Liquid Robotics, a Boeing Company.

Table 1 List of the instruments onboard the Honey Badger with their locations on the Wave Glider (Fig. 2) and their programmed sample frequency.

Sensor (location)	Variables (units)	Interval	Available in near real time?	
Sea-bird scientific’s gpCTD (2)	Water temperature (°C), salinity, density (dBar)	48 h−1	Yes	
Turner Designs’ C3™ submersible fluorometer with antifouling coating (2)	Colored dissolved organic mater (CDOM) (RFU), chlorophyll-a (RFU), and phycoerythrin fluorescence (RFU)	6 h−1	Yes	
Turner Designs’ C3™ submersible fluorometer without antifouling coating (2)	Colored dissolved organic mater (CDOM) (RFU), chlorophyll-a (RFU), and phycoerythrin fluorescence (RFU)	6 h−1	Yes	
AirMar Technology’s WX series ultrasonic WeatherStation® (1)	Air temperature (°C), pressure (mBar), average wind speed (knots) and direction (degrees true)	6 h−1	Yes	
Datawell BV’s MOSE (2)	Significant wave height (m) and direction (degrees true)	2 h−1	Yes	
Cannon G10 camera (2)	Downward facing camera for imaging the sub-body	6 h−1	No	
Turner Designs’ PhytoFlash (4)	Fo, Fm, Fv, yield (Fv:Fm)	6 h−1	Yes	
Sequoia Scientific LISST-Holo (5)	Holographic microscopic images of the water	1 Burst of 15 images every 6 h	No	

The Turner C3 fluorometers were equipped with excitation and emission filters for chl a a, phycoerythrin, and colored dissolved organic material (CDOM) with values reported in fluorescence units. The C3 sensors were deployed on either side of the centerline with a port and starboard sensor. The port C3 sensor and optical port for the look-down camera were coated with a ∼30 μm layer of ClearSignal antifouling compound (Severn Marine Technologies, Annapolis, MD, USA) in spring, 2014. Due to technical difficulties, the mission was delayed a year with unknown effects on the viability of the coating. The look-down camera began recording on July 1, 2015 and imaged vertically below the float for examining the umbilical and glider as needed but also captured images of fish and biofouling over the course of the mission.

The Holo uses collimated laser light to create refraction patterns from particles that are then recorded by camera to create a hologram (Davies et al., 2015). Software provided by Sequoia Scientific Inc. (Holo_Batch v. 3.1) reconstructed multiple holograms into grayscale images. Particle biovolume was calculated based on a cross-section area projected into a sphere. Holo_Detail (v. 3.1) was used to process each hologram in greater detail to identify Hemiaulus and Rhizosolenia spp. Isolated hologram areas could be imaged individually as 0.1–1 mm thick sections allowing detailed images layer by layer. The sampling rate of 15 holographic images (30 s between images) every 6 h was set prior to launch based on worst case power consumption calculations and could not be modified once underway. The 15-image bursts taken every 6 h were combined to form one record yielding four records (bursts) d−1. The Holo sampling volume was 1.86 mL per image with the 15-image burst sampling a total of 27.9 mL. Dye studies prior to the mission indicated the 30 s between images was sufficient for full chamber volume replacement.

The large file size (∼2 MB) of each raw Holo hologram precluded satellite transmission and were only available for analysis after the Honey Badger’s recovery in November 2015. Upon recovery of the drive, 9,336 holographic images were analyzed with the Holo_Batch and Holo_Detail at the University of Texas at Austin’s Marine Science Institute. Comparison of Holo_Batch processing and individual Holo_Detail processing of the same images indicated progressive loss of recognizable diatoms over the mission due to biofouling (examples given in Fig. S1). Therefore, Hemiaulus and Rhizosolenia cells were quantified using the Holo_Detail software on every hologram with distinctive diffraction patterns indicating when particles were present. While using the Holo_Detail to enumerate diatoms was more time-intensive than using the montages of in-focus particles produced by the Holo_Batch, it was necessary as the montages often failed to show Hemiaulus or Rhizosolenia cells when they were clearly identifiable in Holo_Detail. The small size of individual Hemiaulus cells (∼15 μm) and light silicification also contributed to difficulties in using the batch analysis mode as biofouling interference increased.

The Holo’s sampling capability allowed counting cells with a minimum concentration of 36 cells L−1. Individual Hemiaulus cells were at the size threshold of the Holo and hard to differentiate from other small cells unless they were in recognizable chains. In addition, Hemiaulus cells occurred as both individual chains and aggregations of various size. Chains were defined as three or more Hemiaulus cells which formed a curve with clear ends which did not cross itself or others more than once. Aggregates were defined as Hemiaulus cells in a chain or multiple chains with multiple ends or no discernable ends which crossed itself, other chains, or other particles multiple times.

Hologram processing also returned calculated biovolume for all detected particles after calculating their equivalent spherical diameter. The biovolume was automatically separated into bins based on equivalent spherical diameter from 2.5 to 9,847 μm (50 bins with the upper size limit of each bin being 1.18 times the lower limit). The diatoms of interest in this study have an equivalent spherical diameter between 13 and 60 μm so a subset of bins (13.1–58.1 μm) were chosen to focus the analysis. Holograms with schlieren (optical anomalies in transparent mediums), microbubbles or blank images were manually removed from the analyses.

Biofouling interference was removed using the manufacturer’s recommended procedure to average the biovolume over large groups of images. This procedure generated a constant signal that represented a consistent particle presence assumed to be biofouling. We arbitrarily averaged groups of 510 holograms representing an 8.5-day window for a total of 14 background signatures. This signature was subtracted from each hologram in the specified window to generate a biofouling-corrected biovolume. Details of this correction and effects on the result are included as Supplemental Information and Figs. S1 and S2.

Pulse amplitude modulation fluorometry (Schreiber, 2004) determination of Fv:Fm (PhytoFlash sample frequency = 6 samples h−1) was used to evaluate phytoplankton photophysiology. The PhytoFlash sampled at 10 min intervals but was accelerated to 1 min intervals from July 27 to 28 to test the system’s resiliency to increased sampling rates. The port C3 sensor was on a fixed 10 min sampling interval with 10 samples averaged to generate a single value. The starboard sensor was reprogrammable via remote communications and was varied in sampling timing and averaging at various points in the mission. Changes from multi-point averaging to single point reporting resulted in systematic and predictable baseline shifts. The reasons for these changes are unknown. Iron stress was evaluated using the variable fluorescence criteria of Behrenfeld & Milligan (2013) simplified for the lower sampling rate of the PhytoFlash. In a macronutrient limited environment with sufficient iron, the nocturnal Fv:Fm is greater than the diurnal Fv:Fm. In an iron limited environment, the reverse is true. Time averaging (night time average of 36 data points; 08:00–13:59 UTC and daytime average of 54 data points; 18:00–02:59 UTC) was required to obtain a stable signal and timed to avoid the observed crepuscular Fv:Fm excursions. The PhytoFlash shutdown and missed samples at an increasing frequency during the mission and eventually failed completely in early September (traced to software issues). To ensure a comparable day/night sampling, only periods with 75% or more of the expected number of samples were included in the iron-limitation analysis and both periods for a date were required to meet the above standard. These criteria resulted in the removal of 33 of the 94 days of data collected over the mission. The entire Fv:Fm dataset was plotted vs time for a visual inspection of the data as well.

Aqua MODIS satellite’s 8-day composite of daily chl a was used to produce an animation showing the development of the blooms in the NPSG during the 2015 bloom season (June–November) and the position of the Honey Badger’s track (Video S1 at https://figshare.com/articles/S1_movie_mp4/5993644). The raw data from the C3s, gpCTD, AIS, MOSE, PhytoFlash, and weather station are archived at BCO-DMO (http://www.bco-dmo.org/project/505589). The BCO-DMO site also contains the raw holograms, the biovolume data, as well as the Hemiaulus and Rhizosolenia abundance data.

Results

Extensive biofouling on several of the optical windows occurred during the mission. A time series of images from the look down camera illustrates the development over time of barnacles and associated organisms (Fig. S3). A metal incompatibility with internal screw in the LISST-Holo camera system mount resulted in significant corrosion (Fig. S4); however, it did not encroach into the sample plane and no data was lost. Honey Badger collected 5 months of salinity, surface water temperature, diatom abundance, photophysiology, and biovolume data from the NPSG. A nine-point running average (Fig. 3, gray line) and daily average were used to remove changes due to rain events or sensor errors in the gpCTD. The daily averaged water salinity and temperature data (Fig. 3, color-coded by latitude) ranged from 22.8 to 27.8 °C, and 34.6–35.6 salinity. The lower salinity water near Hawai’i is evident at the beginning and ending of the mission. The Honey Badger did not cross the sub-tropical front, which is characterized by salinity ∼34.5 found at ∼30°N (Wilson et al., 2013). The pronounced temperature-salinity gradient from the center of the gyre to Hawai’i is evident in the continuous decrease in salinity and increase in temperature along the straight line transect from the farthest north point (29.245°N and 152.40°W) on September 12, 2015 to just north of Hawai’i on October 23, 2015 (20.67°N and 155.47°W).

Figure 3 Time series of the hydrographic properties from the Honey Badger’s gpCTD sensor.

(A) Salinity. (B) Temperature (°C). The gray lines are the data with a nine-point smoothing, the color-coded dots are daily average values.

The study area underwent a general chl a increase over the course of the mission that was evident visually as a shift from deep blue to light green in mid-July (Video S1). This increase was quantitatively expressed as the average of chl a values from all pixels in the study area (Fig. 4). Following a period of uniformly low chl a concentration throughout the study area in June–July 2015 (Fig. 4), in mid-July chl a concentrations throughout the study area increased concurrent with increased chl a variability (increased standard deviation around the mean) due to chl a blooms (Video S1). This period of elevated bloom activity extended from August 1 to September 15. During this period, there were multiple blooms evident where the satellite chl a exceeded 0.2 mg m−3. The brief decrease in late September was followed by an increase in average chl a through the end of the mission.

Figure 4 Average chl a concentration of the study area over time.

All chl a per pixel values from Aqua MODIS 8-day composite data within the study area (bounded by 19 to 30°N and 157–144°W) were averaged to generate a single daily value for the study area. Solid line = average chl concentration for the study area. Dashed lines = average chl concentration ± 1 s.d.

The two float-mounted Turner C3 fluorometers produced erratic signals and random shifts in baseline values (Fig. 5). The sensors did not parallel each other except for a general increase in the cyanobacteria pigment phycoerythrin from September 21, 2015 to the end of the mission, nor did the satellite chl a values at Honey Badgers location note similar fluctuations. The C3 data sets were excluded from further analysis due to a lack of an independent diagnostic test to determine which data points were reflective of the water properties and which were noise or errors introduced by the sensor.

Figure 5 C3 fluorometer data from the Honey Badger.

Note scale shifts between plots. (A–C) Sensor coated with antifouling compound. (A) Chl. (B) CDOM. (C) Phycoerythrin. (D–F) Uncoated sensor. (D) Chl. (E) CDOM. (F) Phycoerythrin. Study site: RFU, relative fluorescence units.

Hemiaulus and Rhizosolenia cells were readily identifiable in the processed holograms both as chains and aggregates (Fig. 6). Hemiaulus cells were identifiable as either curved or spiral chains (Fig. 6A) as well as aggregates of varying degrees of complexity (Figs. 6B and 6C). With three cells required to define identify a Hemiaulus, the minimum reported concentration is 108 cells L−1. In Rhizosolenia, the symbiont of Richelia intracelluaris was visible as well (Fig. 6D arrows). Mean Hemiaulus abundance over the entire mission was 303 cells L−1 (s.d. = 1.0 × 103 cells L−1, n = 610) and mean Rhizosolenia abundance was 63 cells L−1 (s.d. = 2.7 × 102 cells L−1, n = 610) over all the samples. However, of the 610 samples, only 208 contained Hemiaulus cells and 207 contained Rhizosolenia cells. When present, the average Hemiaulus abundance was 8.9 × 102 L−1 (s.d. = 1.6 × 103 cells L−1, n = 208). Of the samples containing Rhizosolenia cells, the average abundance was 1.8 × 102 cells L−1 (s.d. = 4.5 × 102 cells L−1, n = 207). Hemiaulus maximum abundance in the averaged 15 image burst was 1.4 × 104 cells L−1 on August 2, 2015 and the Rhizosolenia maximum abundance was 2.8 × 103 cells L−1 on August 16, 2015 (Fig. 7). Blooms were defined operationally as occurring when the abundance value was two s.d. above the mean present values, resulting in a threshold of 4 × 103 cells L−1 for Hemiaulus and 1 × 103 cells L−1 for Rhizosolenia.

Figure 6 Processed holographic images of Hemiaulus and Rhizosolenia cells and aggregates.

(A) Curved chain of Hemiaulus hauckii. Each dark dot is the cell mass separated from adjacent cells by siliceous structures. Images have been contrast enhanced for clarity. (B) Hemiaulus aggregate. (C) Hemiaulus aggregate. (D) Two complete Rhizosolenia cells and half of the next cell with their symbionts Richelia (arrows) located at the apex of the cells.

Figure 7 Time series comparisons between the Aqua MODIS a and the in situ data collected by the Honey Badger’s sensors.

(A) 8-day composite data from Aqua MODIS showing surface chl a concentrations (mg m−3) near Honey Badger’s location. (B) Average daily Fm (maximum fluorescence) from the Phytoflash. (C) Hemiaulus abundance (cells L−1). (D) Rhizosolenia abundance (cells L−1). (E) Biovolume from 11–58 μm bins. (F) Average Fv:Fm between 08:00–13:59 UTC (dark-adapted value). Blue and yellow shaded area indicate the Hemiaulus and Rhizosolenia bloom, respectively.

Surface chl a (satellite derived) at Honey Badger’s position underwent a ∼2-fold variation over the mission (Fig. 7A) with a sharp increase on August 2, 2015, followed by considerable day to day patchiness evident throughout the rest of the mission. A similar pattern was seen in the Phytoflash Fm data from sub body at ∼7 m (Fig. 7B) until the data collection failed on September 1, 2015. Two blooms were sampled, a Hemiaulus bloom on August 2–4, 2015 and a Rhizosolenia bloom on August 15–17, 2015. Diatom abundance (Figs. 7C and 7D) was patchy with two order of magnitude changes occurring within between adjacent Holo bursts in the blooms, a distance of approximately 10 km. The Hemiaulus bloom was dominated by Hemiaulus (96% of total diatoms; Fig. 7C) while the Rhizosolenia bloom was dominated by Rhizosolenia (75% of total diatoms; Fig. 7D). However, neither bloom reached the 0.15 mg m−3 chl a threshold used to identify a satellite chl a bloom. The two blooms were separated in space and in time (Figs. 7 and 8) and both had increases in biovolume (Fig. 7E). The larger Rhizosolenia cells contributed nearly 2/3 more biovolume on August 15–17, 2015 despite the cell numbers being only 1/3 that of the Hemiaulus bloom. The satellite chl a signature was still faint when Honey Badger sampled the Hemiaulus bloom from August 2 to 4, 2015 (compare Figs. 8A and 8B) but continued to develop after the Honey Badger left the area (Video S1). The Rhizosolenia bloom sampled by the Honey Badger from August 15 to 17, 2015 did not have a well-defined satellite chl a signal (Figs. 7A, 7C and 8B). However, the PhytoFlash Fm (Fig. 7F) was approximately 33% higher in the Rhizosolenia bloom than the Hemiaulus bloom.

Figure 8 Hemiaulus aggregated and free-living form distribution.

(A) and (B) Aqua MODIS chlorophyll surface concentration and Honey Badger’s position. Black dot = mission track position, red–white–black crosshair = Honey Badger’s position on day of satellite image. (A) August 3, 2015; Hemiaulus bloom. (B) August 16, 2015; Rhizosolenia bloom. (C) Time-series plot of Hemiaulus abundance in the free-living or aggregated form. (D) Locations of non-aggregated Hemiaulus cells and locations of aggregates. Red triangle = aggregate. Circles = non-aggregated Hemiaulus cells L−1, size is proportional to abundance. The green area indicates where Honey Badger sampled during the SEP time window (15 July–15 August).

Two declining blooms evident in the chl a animation were sampled (August 23–25, 2015 and September 14–16, 2015; Video S1). In both cases, no aggregates were seen in the Holo and the maximum local abundance ∼300 cells L−1 was reached in only one burst in each area. The rest of the bursts were devoid of Hemiaulus. However, the lookdown camera imaged what appeared to be a mass occurrence of small white flocs (Fig. S3B). Their identity could not be confirmed, but the size and shape are consistent with either marine aggregates or possibly colonial radiolarians.

Maximum Fv:Fm values (∼0.6) were associated with the Hemiaulus and the Rhizosolenia peak abundance values (Fig. 7E) although the data loss on August 2 may have missed higher Fv:Fm values. During the period of the two blooms (August 2–17), the Fv:Fm values underwent day to day changes in magnitude that were visibly distinct from the period before and after.

The Holo captured 31 Hemiaulus aggregates in 23 sampling bursts (Table 2; Figs. 8C and 8D) out of 610 total bursts over the mission (3.8%) or 11% of samples when any Hemiaulus were present. Aggregates shared common characteristics of curled chains of various sizes tangled together to create a characteristic shape (Fig. 6) and were easily identified when compared to diver-collected aggregates (Fig. S5). When present, 72 ± 25% (s.d., n = 23) of the total Hemiaulus cells were present in aggregated form (Fig. 8C; Table 2). They were not limited to regions where non-aggregated Hemiaulus cells were abundant (Figs. 8C and 8D) and were observed from June 27, 2015 and October 25, 2015 with 13 of the 24 locations outside the time window of the summer export pulse (green shading in Fig. 8D). Within the holograms containing Hemiaulus aggregates, the average number of identifiable aggregated cells was 47 ± 42 (s.d., n = 31) with a minimum of seven (two small crossed chains) and a maximum of 220. Due to the complex 3D structures of some of the aggregates, it is likely that cell counts for aggregates are underestimates.

Table 2 Hemiaulus aggregate locations and contribution to total Hemiaulus abundance.

Aggregate events outside the SEP		Aggregate events within the SEP	
Date (UTC)	n	Location	% Hemiaulus in Aggregates (total cells L−1)	Date (UTC)	n	Location	% Hemiaulus in aggregates (total cells L−1)	
°N	°N	°N	°W	
6/27/15 8:07	1	28.41	154.45	66.6 (861)	7/20/15 17:14	1	26.25	147.68	82.9 (1,471)	
7/10/15 1:06	1	26.69	151.96	91.8 (1,757)	7/31/15 15:31	1	25.80	145.01	95.0 (2,869)	
8/17/15 4:01	1	25.10	151.45	78.6 (1,506)	8/02/15 10:09	3	25.24	145.52	42.0 (13,737)	
8/18/15 16:37	1	25.15	152.15	15.5 (2,080)	8/03/15 4:29	1	25.00	145.71	29.7 (4,232)	
8/19/15 10:55	1	25.18	152.51	78.6 (1,506)	8/03/15 10:38	3	24.93	145.76	56.0 (5,702)	
9/2/15 16:29	1	27.44	153.74	64.7 (1,219)	8/05/15 17:34	2	24.71	146.65	85.1 (2,403)	
9/27/15 14:14	1	26.12	153.58	100.0 (2,618)	8/08/15 0:22	1	24.81	147.83	80.7 (3,156)	
10/15/15 21:22	1	21.80	155.08	95.5 (2,367)	8/08/15 6:31	2	24.81	147.93	100.0 (3,802)	
10/20/15 23:22	2	20.81	155.46	69.3 (2,690)	8/09/15 19:10	1	24.86	148.50	39.0 (2,116)	
10/21/15 5:28	1	20.77	155.47	31.8 (789)	8/10/15 1:10	1	24.87	148.57	75.8 (2,367)	
10/25/15 0:58	2	20.54	155.18	90.1 (2,546)						
10/25/15 7:04	1	20.56	155.11	95.7 (8,249)						
10/25/15 13:10	1	20.54	155.04	100.0 (2,009)						
Notes:

The Hemiaulus aggregate events outside (left) and within (right) the 15 July–15 August summer export pulse. N = number of aggregates in each 15-image burst at that location. The Hemiaulus bloom event is in bold, italicized text.

A single aggregate in a 15-image burst represents, on average, 36 aggregates L−1. Maximum abundance was present during the Hemiaulus bloom (August 2–4, 2015) where normalized abundance was 108 aggregates L−1. The highest sustained aggregate abundance was during the early August bloom when aggregates were observed in six of nine successive days (Table 2). However, there was no significant relationship between aggregated and non-aggregated cell abundance (r2 = 0.12, p = 0.5, n = 23) overall in the data set. On 3 of the 23 bursts where aggregates were observed, they were the only form of Hemiaulus present.

The Fv:Fm values underwent diel excursions typical of high-light populations experiencing solar-induced photoinhibition and down-regulation of photosynthetic activity where yields were greatest in the dark period and lower during the daytime (Fig. 9A). Crepuscular excursions were evident in many, but not all diel rhythms. From visual inspection of the entire mission dataset, there was no reversal of the diel rhythm suggestive of Fe-stress. The quantitative diurnal:nocturnal Fv:Fm ratio remained positive indicative of a macro-nutrient limited environment (Fig. 9B) although there was a long-term downward slope. The near zero values after September 1, 2015 were the result of compromised PhytoFlash data as the Fo and Fm values simultaneously drifted upwards resulting in loss of Fv:Fm (details in Fig. S6). August 31, 2015 was the last date with uncompromised data before the PhytoFlash completely shutdown on September 9, 2015.

Figure 9 PhytoFlash Fv:Fm diel rhythm sample and iron limitation index.

(A) Sample of the typical diel rhythm observed in the PhytoFlash Fv:Fm measurements. The signal is down-regulated during the daytime and returns to the maximum value during the dark period while macro-nutrient limited. Dark bars = 08:00–13:59 UTC (nocturnal period used in the calculation). Light bars = 18:00–02:59 UTC (diurnal period used in the calculation). (B) Time-series of the dark-averaged Fv:Fm minus the light-averaged Fv:Fm. Red points indicate where the sample number did not meet the threshold for calculation (see Methods). Asterisks are points where the data was compromised (see Text S1).

Discussion

The Wave Glider SV2 as a sampling platform

The Wave Glider SV2 Honey Badger successfully returned from a 5-month mission with all sensors undamaged. All sensors reported data, although at varying frequency and reliability, throughout the mission with the exception of the PhytoFlash. As a prototype mission, it was successful at deploying and recovering optical and imaging sensors specific to phytoplankton research questions. Individual sensors suffered from degradation associated with either platform computer software issues (PhytoFlash) or environmental biofouling (C3s and the LISST-Holo).

Post-mission inspection by Turner Designs indicated the PhytoFlash operated properly when removed from the glider, suggesting the system interface with the glider had failed. The SV2 was the first production model of Wave Gliders. The customized software used to power and communicate with the PhytoFlash was not part of the original system’s dedicated software and gradually created insurmountable conflicts that led eventually to a complete failure. The newer generation (SV3) has a more robust on-board computer interface more amenable to customization and this is not likely to be a future issue.

One of the goals of the mission was to sample regions with chl a concentrations >0.15 mg m−3. The waypoints for glider were partially chosen based on the Aqua MODIS’s chl a data. Daily images were often incomplete due to cloud cover as well as being outside the daily imaging path. The 8-day composite of the Aqua MODIS satellite data provided a more complete image of the regional chl a concentrations, however, the 8-day images used for daily decision making on the glider’s movements were based on data that may have been up to 4-days-old. This delay resulted in a few missed sampling opportunities (Video S1) since chl a maps of the region the data were incomplete as waypoints were determined. This was particularly evident in the August Hemiaulus bloom. The magnitude of the bloom was not evident in the satellite imagery until the glider was a week past it and nearly halfway to a developing bloom to the west.

Biological observations

During the June–November timeframe of this mission, Hemiaulus and Rhizosolenia were the dominant diatom genera observed by the Holo in the NPSG chl a blooms, reaffirming Guillard & Kilham’s (1977) characterization of these taxa as persistent diatom representatives of the oligotrophic open ocean flora. The Holo’s resolution limit (∼15 μm) could not image the smaller pennate diatoms such as Mastogloia that frequently co-dominate in these blooms. The Hemiaulus abundance (104 cells L−1) noted in the August 2–4, 2015 bloom is consistent with previous reports of open Pacific Ocean blooms where Mastogloia is a co-dominant (Brzezinski, Villareal & Lipschultz, 1998; Scharek et al., 1999; Venrick, 1974; Villareal et al., 2012). Thus, it is probable that additional diatoms were present and contributing to the satellite chl a signature.

The patchiness in the abundance of Hemiaulus and the Rhizosolenia symbiosis was unexpected. Approximately 2/3 of the bursts contained neither of these taxa. In some cases, the next sampling burst (6 h later, or approximately 10 km) would observe ∼103–104 cells L−1. Such variation has been noted before from discrete ship sampling stations (Fong et al., 2008; Venrick, 1974; Villareal et al., 2012) but with little ability to sustain 6 h sampling intervals for months. The most extreme gradients were associated with developing blooms suggesting that the factors driving blooms are highly localized and not represented by the average nutrient or hydrographic characteristics. Calil et al. (2011) reported satellite chl a features in this gyre developed rapidly at frontal interfaces between mesoscale features as the result of sub-mesoscale ageostrophic flows resulting in transient up and down welling. This spatial development scale is consistent with the abundance increase noted in the two observed diatom blooms and warrants further investigation into the role that mesoscale frontal features play in diatom-diazotroph association dynamics. However, there are no mechanisms suggested to address the variability in the background concentrations (101–102 cell L−1) of these taxa presumably adapted to uniformly oligotrophic conditions.

Unlike previous studies using settled water samples or nets, we were able to record and partition Hemiaulus into aggregated or unaggregated abundance. Hemiaulus aggregates (Villareal et al., 2011) occurred throughout the mission, even in regions of low non-aggregated Hemiaulus abundance (Figs. 7 and 9). The presence of one or more aggregates usually dominated the total abundance (Table 2) and on three occasions represented the entire Hemiaulus biomass seen. Maximum abundance (108 aggregates L−1) and highest sustained aggregate abundance were both present during the Hemiaulus bloom (August 2–4, 2015) where aggregated Hemiaulus represented 29–56% of the total Hemiaulus present in the bursts.

With an aggregate occurrence in 11% of the samples containing Hemiaulus, we examine what principles of diatom aggregation are relevant in this environment. Jackson’s general coagulation model for diatom aggregates (Jackson, 1990a, 1990b) suggest senescence, elevated concentrations, and enhanced stickiness play a key role in aggregation formation. In our data, aggregate density was highest in the August bloom, consistent with this model. However, the long chains and elevated Fv:Fm suggests a rapidly growing Hemiaulus population and the continued increase in the bloom area chl a after Honey Badger departed (Video S1) suggests that this bloom was sampled early in its development. The aggregated form dominated total abundance when present, and aggregates appeared largely monospecific, at least within the resolution limits of the Holo. In contrast, diatom aggregates in coastal waters scavenge other particles and can sweep the water clear as they sink (Alldredge & Gotschalk, 1989, 1990; Alldredge & Silver, 1988). We suggest that aggregated forms of Hemiaulus are not solely the result of high rates of collision and sticking between Hemiaulus cell. Much like Rhizosolenia mats (Villareal & Carpenter, 1989), they may be a natural growth form of Hemiaulus that results from curled chains twisting back on themselves. Further collisions may play a role but appear unlikely in the low density conditions that generally prevailed in this study.

Combined diver and net collections in 2003 (Villareal et al., 2011) found high Hemiaulus abundance was coupled to an aggregation snowstorm (Fig. S5) and allows us to examine whether the Holo’s aggregate abundance data is credible. Using net-collected abundance data from the 2003 bloom (maximum abundance: 2,500 cells L−3) and our average cells per aggregates in this study (47 cells), we calculate a potential for ∼50 aggregates L−1 for the 2003 Hemiaulus snowstorm. The aggregates visible to divers (centimeter-sized) are substantially larger than the aggregates observed by the Holo (millimeter-sized), so this is likely an overestimate of abundance in the 2003 snowstorm. However, the value is similar to the detection limit represented by one aggregate per 15 image Holo burst (36 aggregates L−1) and suggests the Holo data are the correct order of magnitude. Combined with the high proportion of samples containing aggregates (11%), our limited sample volume (∼28 mL), the broad aggregate distribution, and the lack of a satellite signature from the 2003 snowstorm (Villareal et al., 2011), we conclude that dense Hemiaulus aggregation events are more common than reported. Pilskaln et al. (2005) reported marine snow aggregates on the order of 1–10 L−1 at 28–30°N along a transect from HI to CA suggesting that Hemiaulus aggregates are part of rich collection of macroscopic particles rarely sampled. The incidental observation from the lookdown camera in a fading bloom of what appeared to be large aggregates in a fading bloom were at too low a density to be sampled by the Holo, but sufficiently large to be visible to the camera (Fig. S3B). Multiple imaging technologies on the vehicle are clearly needed to further detail this type of event.

Regularly occurring Hemiaulus aggregates could be an important food source to organisms in the open ocean due to their high concentration of carbon and nitrogen. They could also play an important role in the global carbon cycle since aggregated forms, when physiologically stressed, tend to sink much faster than non-aggregated particles (Stemmann & Boss, 2012) and can scavenge other suspended particles as they sink to depth (Alldredge & Silver, 1988). Station ALOHA sediment trap data indicated that during the 13-year record, the summer export pulse resulted in ∼20% of the annual carbon export to the benthos at >5,000 m (Karl et al., 2012) with high sinking rates (102 m d−1) requiring aggregates as a dominant mode of transport (Scharek et al., 1999; Scharek, Tupas & Karl, 1999). Our data show that Hemiaulus aggregates extend deep into the North Pacific gyre and support the idea that the role of the summer export pulse may be much wider than Station ALOHA waters near Hawai’i. However, there is no evidence that aggregate formation, per se, is linked to the hypothesized annual rhythm driving the summer export pulse. They occur independently of the summer export pulse.

We found no evidence of iron limitation during our sampling with the caveat that the PhytoFlash measures a property of the phytoplankton community present, not a diatom-diazotroph association specific stress. However, even during the Hemiaulus and Rhizosolenia blooms observed on August 3, 2015 and August 16, 2015, the Fe index did not suggest iron limitation or iron stress. From June 1, 2015 to August 31, 2015, the dark-averaged Fv:Fm stayed above the light-averaged values, agreeing with the 2006 study by Behrenfeld et al. (2006) which classified this area as having a type I regime with low macronutrients but sufficient iron supplies.

Conclusions

The Honey Badger offered a unique look into the remote oligotrophic NPSG during its 5-month, 5,690 km mission. While some of the sensors failed during the mission (PhytoFlash) or produced uninterpretable data (C3s), the mission was a success in that other sensors (LISST-Holo) recorded novel data over an extensive period of time (5 months) and wide geographic extent, and the glider returned intact. The Honey Badger and its sensors allowed for a persistent presence in the NPSG during the late summer/early fall bloom season.

The long-term deployment of both imaging and photosynthetic efficiency sensors on a mobile sampling platform provided novel information on the composition and physiology of remote diatom blooms. The region showed no evidence of iron limitation despite the presence of DDAs at 104 concentrations. Hemiaulus aggregates were widespread and observed outside the July 15–August 15 summer export pulse (Karl et al., 2012) window suggesting that the predictable timing of the summer export pulse cannot be uniquely attributed to a rhythm in aggregate formation. If aggregates are consistent vectors for vertical transport at some stage, then the potential for a basin-wide summer export pulse is enhanced. When present, Hemiaulus aggregates are abundant (>10 L−1) and dominate the total Hemiaulus present. Their general characteristics are distinct from coastal diatom aggregates and more similar to Rhizosolenia mats (Alldredge & Silver, 1982; Carpenter et al., 1977; Villareal et al., 2014), suggesting Hemiaulus aggregates are a natural growth form. Their broad and persistent occurrence suggests they do not have consistently high sinking rates. The PhytoFlash and the Holo data are generally uncoupled from the satellite chl a concentrations which illustrates the added value of in situ sampling to understand the community structure and photophysiological characteristics of these blooms in remote open ocean habitats.

Supplemental Information

Supplemental Information 1 Biofouling correction for the LISST-Holo.

Click here for additional data file.

Supplemental Information 2 Fig. S1. Hologram fouling comparison.

(A) clean hologram early in mission (image 004-0973; 6/14/15 2:54), (B) dirty hologram late in mission (image 004-9029), note the extensive background refraction patterns caused by fouling. (C) Batch processed (image 004-0973: 10/27/15 19:57) (D) Batch processed (image 004-0973; 6/14/15 02:54).

Click here for additional data file.

Supplemental Information 3 Fig. S2. Biofouling corrections and impact on volume calculations.

(A) Total biovolume reported by the LISST-Holo software (grey line) and corrected biovolume after background removal (black line). (B) Percentage of biovolume removed from the images over time. Vertical dotted lines are endpoints of the background averaging window (see text for details).

Click here for additional data file.

Supplemental Information 4 Fig. S3. Images from the Canon G10 downward facing camera illustrating biofouling buildup.

(A) IMG_20150701220016, 1 July 2015 1200 local time, (B) IMG_20150823220014, 23 Aug. 2015 1200 local time. This image illustrates an apparent accumulation of marine aggregates. (C) IMG 20150920170016. 21 Sept. 2015 0700 local time. (D) IMG_201510312200 31 Oct. 2015 1200 local time.

Click here for additional data file.

Supplemental Information 5 Fig. S4. LISST-Holo sampling chamber and camera.

(A) Looking into the sampling chamber. Corrosion is evident as a yellow-white material on the lower left of the camera. Note that it does not extend into the active optical sampling area. No macroscopic fouling is evident. (B) close up of corrosion after LISST-Holo was removed from the tow body. (C) Laser optical window noting no corrosion or macroscopic fouling.

Click here for additional data file.

Supplemental Information 6 Fig. S5. Hemiaulus snowstorm observed by divers on 31 Aug. 2003 in the central N. Pacific gyre.

(A) The arrow indicates a Rhizosolenia mat. The remainder of the flocs were Hemiaulus aggregates (B) Stereoscope microgaphy of a Hemiaulus aggregate. Figure from Villareal et al. (2011).

Click here for additional data file.

Supplemental Information 7 Fig. S6. PhytoFlash diel rhythms immediately before the PhytoFlash failure in early September.

Fv:Fmvalues showed a normal diel rhythm (nocturnal maxima; diurnal minima) up to 1 September 2015. At this time, both the Fmand Fo used to calculated this value began a fatal upward drift that was not reflective of ambient chlorophyll concentrations from the satellite data. The values exceeded previously observed values by a factor of 3–4 until the instrument shut down and could not be restarted. Data after mid-day 1 Sept. is considered compromised and not useable for Fe-index determination. It is indicated on Fig. 9b as the asterisked points.

Click here for additional data file.

Supplemental Information 8 Supplemental Figure Legends for download.

Click here for additional data file.

We wish to thank Liquid Robotics, a Boeing company, for providing the glider time as part of the PacX Challenge award and our project manager Danny Merritt for his contributions to the mission success. We acknowledge John Walpert (GERG) for adapting some of equipment to the SV2. In addition, the skilled field testing and support provided by Brad Woolhiser, Chuck Shaver, Dustin Boettcher, and Vas Podorean at the LR test facility in Kawaihae, HI is gratefully acknowledged. We wish to thank Bob Simons (SWFSC/ERD) for putting the Honey Badger data on ERDDAP and Lynn DeWitt (SWFSC/ERD) for creating the project website (http://oceanview.pfeg.noaa.gov/MAGI).

Additional Information and Declarations

Competing Interests

Author Contributions

Data Availability

The authors declare that they have no competing interests. Liquid Robotics provided a cash grant, 6 months of Wave Glider time, and technical assistance for this project. They have not seen nor commented on the data or the manuscript.

Emily E. Anderson performed the experiments, analyzed the data, prepared figures and/or tables, authored or reviewed drafts of the paper, approved the final draft.

Cara Wilson conceived and designed the experiments, performed the experiments, analyzed the data, contributed reagents/materials/analysis tools, prepared figures and/or tables, authored or reviewed drafts of the paper, approved the final draft.

Anthony H. Knap contributed reagents/materials/analysis tools, authored or reviewed drafts of the paper, approved the final draft, provided expertise in integrating sensors into the vehicle.

Tracy A. Villareal conceived and designed the experiments, performed the experiments, analyzed the data, contributed reagents/materials/analysis tools, prepared figures and/or tables, authored or reviewed drafts of the paper, approved the final draft.

The following information was supplied regarding data availability:

Biological and Chemical Oceanography Data Management Office (BCO-DMO): https://www.bco-dmo.org/project/505589.

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
