# Peer review of "Summer diatom blooms in the eastern North Pacific gyre investigated with a long-endurance autonomous surface vehicle"

_PeerJ, doi:10.7717/peerj.5387_

## Round 0.1 · original submission · Minor Revisions

Both reviewers were very positive about the manuscript. It is clear, reads well and makes an important methodological contribution as well as providing novel observations.

Reviewer 2 has some good suggestions that should be considered in a revision. I agree with Reviewer 2 that there is some prospect for confusion between comments concerning Honey Badger (the specific, named AUV used) and the generic Wave Glider SV2 as a model.

Please address these on a point-by-point basis, explaining the changes you have made, or in your rationale in cases where you disagree with the suggestion.

I have a couple additional comments:

1) Throughout the manuscript “chlorophyll” or “chlorophyll-a” (in Figures) should be replaced with “chlorophyll a”, with the ‘a’ italicized, and the abbreviation “chl” with “chl a”. These are clearer, more conventional, and (I believe) what is meant in all cases.

2) As noted by reviewer 2, in the abstract, the abbreviation “SEP” needs to be defined. It looks like this stands for ‘summer export pulse’; it may be more efficient to simply explain the process in the abstract, since the abbreviation is used only once.

3) Line 60: the orthophosphate ion carries a 3-, not a 2- charge

4) As noted by Reviewer 2, in line 118: insert units after depth (i.e. ‘m’). Be consistent with use of commas for placeholders in numbers, cf. lines 118, 228 and 461.

5) Is it worth stating “return[ing] the glider safely” as an objective? I appreciate that’s always what we hope for, but perhaps something more ambitious/useful like ‘operating for a period of x months with all sensors collecting data’? This also tallies better with the first lines of your discussion.

6) Line 241-2: “phytoplankton physiological health” is vague and imprecise. How about “photosynthetic efficiency”?

7) Statistical results are presented inconsistently. (e.g. lines 354, 359, 367; “s.d.”, “std. dev.”, “n” given or not).

8) It doesn’t seem reasonable to describe the PhytoFlash measurements as a “bulk water property” (line 491). They constitute a community-level measurement of photosynthetic efficiency of Photosystem II, and that’s certainly got issues for discerning cell-specific differences (cf. Franklin et al. 2009. Effect of dead phytoplankton cells on the apparent efficiency of photosystem II. Mar Ecol Prog Ser 382: 35-40).

9) Line 512: does “10+” mean “> 10”? The latter is probably clearer.

10) A somewhat pedantic point, but contrary to many bad examples in the literature, (and as Frank Millero has been reminding us for years- see e.g. Millero et al. 2008. Deep-Sea Research I 55: 50-72) salinity measured by induction-based salinometers is a dimensionless ratio, so “PSU” is nonsensical. (Figure 3, Table 1).

11) In terms of readability, a couple acronyms (SEP, DDA) that don’t save much space are used infrequently enough that they require a reader to flip back and forth to recall what they stand for. The manuscript isn’t space limited…what not simply write them out and save us some trouble.

Reviewer 1 ·

Basic reporting

The paper is well-written. There are a lot of figures and supplemental materials but they all add to the story.

Experimental design

Given the intent (to deploy an autonomous vehicle to examine blooms in the SEP) the experimental design is well-justified. Autonomous platforms have inherent limitations, in that one cannot modify the data collection in real-time very easily, making the resulting analysis focus more heavily on data reduction and QA/QC. This is one of the main points of the paper--how do we use these platforms? I think the authors did a good job of providing both the pros and cons.

Validity of the findings

This manuscript describes a unique combination of autonomous platform and instruments. The authors have previous published on the use of the LRI Wave Glider platform for extended deployments, but that was with a more “traditional” suite of instruments. Here the inclusion of a LISST HOLO provides a unique new dimension. Key aspects of the paper cover both the technical strengths/limitations/issues with these platforms, and apply the data to examine questions about the importance of Hemiaulus in formation of blooms in the SEP.

The technical side of the manuscript provides a nice description of how these new platforms can be used, focusing in particular on data quality and the effect of fouling. The use of the LISST is quite intriguing, and the authors show how taking “bursts” of data can answer ecological questions while managing the considerable data volume produced. Again, the description of how the data were analyzed is useful in and of itself.

The science questions are a bit more fuzzy, with a lot of suggestions and directions for future research. For example, the authors speculate about the role of aggregation, and the potential for submesoscale interactions to promote blooms. This is all reasonable and interesting, but given the available data it must remain speculative. Despite that, some key conclusions emerge:

- there appears to be a background concentration of cells identified by the LISST
- early bloom formation is tied to small (regional) scale variability, and is very patchy—suggesting that autonomous observations are one way to get a better handle on the distribution and initiation of bloom events.
- aggregate formation is likely important

Taken together the combination of technical advances (included data reduction and analysis), and the more speculative assessment of the ecophysiology make this wirth publishing. There are a few specific items that should be cleaned up (see below). Beyond that, the one area that could be strengthened is the analysis of how cell density/aggregation is related to the physical data. It is suggested that ageostrophic flow may result in localized upwelling/downwelling, for example—is there any evidence for that in the other data (such as TS)? A more rigorous analysis would move the discussion into a more quantitative assessment of the role of small-scale variability.


Abstract: SEP is not defined

Lines 102-104: this seems a bit convoluted, in that blooms >10E4 cells/L extend well north at abundance up to 10E4 L-1. Is it necessary to give the abundance twice? And (minor point), greater than is not the same as up to. If this is listed twice because of conflicting reports as to the concentration, then rewrite.

118: add meters (2000-4000m)

154: the tether is variable in length, so it would perhaps be more appropriate to say for this particular configuration it was 7m.

183: not to be picky, but the configuration is based on both excitation and emission, so I imagine the LEDs (excitation) were also chosen for specific wavelengths, i.e. there was no white light source with filters.

382: While Honey Badger was the specific vehicle I suggest referencing WaveGliders more generically, since the emphasis is on what this class of vehicle can accomplish, not the specific vehicle.

Figure 4: the dashed lines are the SD, not the average. And remove the “v” at the end.

Figure5: define RA (running average?) and how you calculated it

supplemental information: shouldn’t there be specific figure captions for each of the supplemental figures?

Additional comments

I'd like to see this published, it is a unique dataset that demonstrates the potential of these platforms when some creativity in instrumentation and data analysis is used.

Reviewer 2 ·

Basic reporting

See below

Experimental design

See below

Validity of the findings

See below

Additional comments

The manuscript by Anderson et al entitled “Summer diatom blooms in the eastern North Pacific gyre investigated with a long-endurance autonomous surface vehicle” (submission 28039) provides a descriptive account of spatial-temporal variations in phytoplankton properties in a region north of the Hawaiian Islands. Data were collected from June to November 2015. A particular interest in this study was the abundance of Hemiaulus and Rhizosolenia cells/aggregates. The authors find these diatoms persistent in the study area and the occurrence of aggregates to span beyond the classic July-August summer export pulse (SEP) period, a finding which has implications on the role of these aggregates on carbon export and raises questions on whether aggregation is a consequence of physical processes or a normal life form. I have to admit that I found this manuscript to be exceptionally well written and thorough. The analyses are robust, limitations and challenges in the measurements are clearly articulated, and conclusions are soundly supported by the observations.

It is extremely rare for me to conduct a manuscript review and not have substantial comments to give, but in this case I have no concerns or criticisms and believe the manuscript is ready to proceed to publication. I would like to congratulate the authors on a job well done.

---

## Round 0.2 · accepted · Accept

Nicely done. All the points raised have been addressed and ambiguities clarified. I caught one additional typo: line 65 "Dizaotrophy".

#